# Novel Coumarin 7-Carboxamide/Sulfonamide Derivatives as Potential Fungicidal Agents: Design, Synthesis, and Biological Evaluation

**DOI:** 10.3390/molecules27206904

**Published:** 2022-10-14

**Authors:** Shu-Guang Zhang, Yu-Qiang Wan, Ya Wen, Wei-Hua Zhang

**Affiliations:** Jiangsu Key Laboratory of Pesticide Science, College of Sciences, Nanjing Agricultural University, Nanjing 210095, China

**Keywords:** coumarin, carboxamide, sulfonamide, fungicidal activity

## Abstract

Coumarin compounds have a variety of biological activities such as anti-tumor, anti-coagulation, anti-HIV, anti-fungal, and insecticidal. Amide and sulfonamide compounds have been used as fungicides for half a century, and dozens of varieties have been developed so far. This study focused on the introduction of carboxamide and sulfonamide moieties in a coumarin core to discover novel derivatives. Based on this strategy, we synthesized two series of novel carboxamide and sulfonamide substituted coumarin derivatives, and their fungicidal activity was also investigated. Some designed compounds possessed potential activities against six phytopathogenic fungi in the primary assays, highlighted by compound **6r**. Compound **6r** exhibited stronger fungicidal activity against *Botrytis cinerea* (EC_50_ = 20.52 µg/mL) and will be the lead structure for further study.

## 1. Introduction

Agricultural diseases are an important factor causing food crises, and plant pathogenic fungi and bacteria cause about 2/3 losses. Fungicides have become an important measure for comprehensively controlling agricultural diseases [1]. However, long-term irregular use of fungicides has resulted in resistance to many fungal diseases and serious harm to the natural environment. Therefore, developing new, environmentally friendly, efficient, and selective green fungicides is of great significance to ensure the safety of food, crops, and ecological environment and promote the sustainable development of agriculture [2,3]. Natural products provide a large number of synthetic templates for the development of new pharmaceutical and pesticide-active molecules due to their novel structure, unique mode of action, and environmental friendliness [4].

Coumarin and its derivatives are widely distributed in natural plants, such as *Rutaceae*, *Leguminosae*, *Umbelliferae*, and *Compositae* [5]. Coumarin derivatives display a variety of biological activities, such as anti-virus, anticancer, antioxidant, antimicrobial, and herbicidal activity. They have been found in widespread application in the fields of pesticides and medicine [6,7,8,9]. As a secondary metabolite of phenylpropanoids synthesized by plants, coumarins have good environmental compatibility, unique physiological activity, and structural plasticity. In the agricultural field, coumarin and its derivatives have many important biological activities. For example, many pathogens can induce plants to secrete coumarin compounds, which can prevent the invasion of pathogens at the infection site and play antibacterial roles in vitro. They are a class of phytoalexins and special lead compounds for the synthesis of small molecules of agriculturally active organic compounds, which has attracted the interest of many pesticide chemists [10]. As a commercial coumarin fungicide, Osthole has a good inhibitory effect on *Fusarium graminearum* and *Phytophthora capsici* and has a good inhibitory effect on melon powdery mildew [11]. Coumoxystrobin containing a coumarin skeleton shows broad-spectrum fungicidal activity against many plant diseases and has been observed in field trials against cucumber downy mildew with a good inhibitory effect [12] (Figure 1).

Inspired by these facts and using Osthole as a lead structure, our group designed and synthesized a series of coumarin derivatives with high fungicidal activity. Compounds such as pyrano[3,2-c]chromene-2,5-diones, pyrrole/pyrazole-substituted coumairn derivatives, furo[3,2-c]coumarins, coumarin-3-carboxamide derivatives, and fluorinated 7-hydroxycoumarin derivatives containing an oxime ether moiety were studied [13,14,15,16,17]. However, their potency to be used as agricultural fungicide still has a long way to go.

Carboxamide and sulfonamide moieties exhibit many important biological activities in the agricultural field. Carboxamide fungicides are a kind of ancient fungicides, the number of which occupies a large proportion. Bayer, BASF, Syngenta, and other companies have successively developed novel varieties of commercial amide fungicides, such as sopyrazam, fluopyram, and bixafen [18,19,20]. Sulfonamide compounds have various physiological activities and are widely used in pesticides. Amisulbrom, which features a sulfonamide group, was applied to control phytophthora and downy mildew [21]. 4-Hydroxycoumarin sulfonamides hybrids show significant antifungal activity in vitro against *M. canis* [22] (Figure 2).

Based on these encouraging results, we considered that combining the coumarin skeleton with carboxamide and sulfonamides moieties may result in diversified coumarin hybrids with high fungicidal activity (Figure 3). Herein, we synthesized a novel series of coumarin 7-carboxamide/sulfonamide derivatives, which involved the Pechmann reaction, carbamate deprotection, and amidation/sulfonamidation. In addition, we examined their fungicidal activity against six phytopathogenic fungi to further improve their fungicidal effects.

## 2. Results and Discussion

### 2.1. Synthesis

The synthesis of coumarin 7-amide/sulfonamide derivatives is depicted in Figure 1. The reported route led to six kinds of 7-aminocoumarin derivatives and involved the Pechmann reaction and carbamate deprotection [23,24]. Five types of aroyl chloride were prepared by the reaction of thionyl chloride on the corresponding aromatic carboxylic acids in CH_2_Cl_2_ under reflux. Five kinds of phenylsulfonyl chlorides were commercially available. The target molecules **5** were synthesized by the reaction between compounds **4** and aroyl chloride in good yields in CH_2_Cl_2_ as the solvent and triethylamine as the base, as shown in Figure 1. It is worth noting that compounds **5k**–**5v** did not form under these reaction conditions. When THF was the solvent and NaHCO_3_ was the base, compounds **5k**–**5v** were obtained in moderate yield. In the synthesis of coumarin 7-sulfonamide derivatives, pyridine was used as the base, CH_2_Cl_2_ as the solvent, and DMAP as the catalyst, and compounds **6a**–**6r** were synthesized subsequently. We also tried to use different kinds of condensation agents to prepare amides. However, due to the poor reactivity of aromatic amines with heterocyclic carboxylic acids, the reaction was not ideal. Finally, carboxylic acid was prepared into a more reactive acyl chloride to react with the amino group. All reaction processes were monitored by thin-layer chromatography (TLC).

The structures of the target molecules listed in Table 1 and Table 2 were confirmed by HRMS, ^1^H NMR, and ^13^C NMR. The data of target compound **5a** was analyzed as a representative example. The ^1^H NMR signal of 10.58 ppm was assigned to the NH of the amide group at 2.34 and 2.06 ppm, the signals corresponding to the methyl groups appeared at positions 4 and 3, respectively. In the ^13^C NMR spectrum, the signal peak at 162.51 ppm confirmed that compound **5a** was assigned to the carbonyl group in amide moiety, and 156.26 ppm indicated the existence of the carbonyl group in the coumarin motif. Two signals at 15.13 and 13.49 ppm were assigned to the methyl groups. In the HRMS spectrum of compound **5a**, the value of the [M + H]^+^ ion absorption signal was 284.0924, which was consistent with the calculated value (284.0917) for C_16_H_13_NO_4_^+^[M + H]^+^.

### 2.2. Antifungal Activity of the Target Molecules

The antifungal activity of the target molecules on plant pathogens was tested by a mycelium growth rate method, and the assay results are shown in Table 3. Osthole was adopted as the positive control fungicide during the assay process. Plant pathogens were from the Department of Pathology, College of Plant Protection, Nanjing Agricultural University. The phytopathogenic fungi included *Botrytis cinerea*, *Alternaria solani*, *Gibberella zeae*, *Rhizoctorzia solani*, *Cucumber anthrax*, and *Alternaria* leaf spot. The target compounds were dissolved in DMSO to generate a stock solution. The compounds possessing good activity (inhibitory rate >60% at 50 μg/mL) were further evaluated using different concentrations by diluting the above solution. DMSO served as the negative control. In general, most of the synthesized compounds exhibited more antifungal activity against *Botrytis cinerea* and *Rhizoctorzia solani* than against *Alternaria solani*, *Gibberella zeae*, *Cucumber anthrax*, and *Alternaria* leaf spot. Noteworthy, four compounds—**6d**, **6e**, **6q**, and **6r**—showed relatively effective control against *Botrytis cinerea* and corresponding inhibition rates of 71.1, 68.2, 78.2, and 81.6%, respectively, which were equivalent to that of the positive control fungicide Osthole (81.1%).

To comprehensively study the antifungal activity of the effective molecules, we further examined the EC_50_ values of these compounds together with Osthole. As shown in Table 4, four compounds (**6d**, **6e**, **6q**, and **6r**) showed fair to good activity; the EC_50_ values were lower than that of Osthole (30.67 μg/mL) against *Botrytis cinerea*. It was worth noting that compound **6r** displayed much more activity than the control Osthole did.

### 2.3. Structure–Activity Relationships

Although the antifungal activity of most coumarin 7-amide/sulfonamide derivatives was poor, making it difficult to extract a clear structure–activity relationship summary, some preliminary conclusions can still be drawn. Firstly, most of the synthesized coumarin derivatives were relatively more active against *Botrytis cinerea* and *Rhizoctorzia solani*, but they usually lacked potency against the other tested fungi. Secondly, the overall effect on antifungal activity of the sulfonamide group seemed helpful compared with that of the carboxamide group. Thirdly, compounds **6d**, **6e**, **6q**, and **6r** exhibited significant antifungal activity against *Botrytis cinerea*, equivalent to that of the positive control fungicide Osthole, and **6r** was identified as the most promising candidate. The results favor the introduction of CH_3_/Cl at the C-3 cite in the coumarin core. Meanwhile, para-Cl/Br substituted to the phenyl ring was helpful to improving the antifungal activity of these molecules.

## 3. Materials and Methods

### 3.1. Chemicals and Instruments

All chemical reagents were purchased from commercial sources used without further purification. The progress of reactions and the purity of products were monitored by TLC using silica gel GF/UV 254. The melting points of coumarin compounds were measured on an X-4 apparatus (uncorrected). ^1^H-NMR and ^13^C-NMR spectra were detected on a Bruker Avance 400 MHz spectrometer (Billerica, MA, USA) with TMS as an internal standard. HR-MS (ESI) spectra were operated using a Thermo Exactive spectrometer (Waltham, MA, USA).

### 3.2. Chemistry

#### 3.2.1. General Procedure for the Synthesis of the Intermediates **4**

7-Aminocoumarins were synthesized through procedures reported in [16,25].

#### 3.2.2. General Procedure for the Synthesis of the Intermediates **5a**–**5j**

Compound **4** (2.5 mmol) was dissolved in dichloromethane (20.0 mL), and then triethylamine (20.0 mmol) was added to the solution and cooled to 0 °C. To the mixture, 5.0 mmol of aroyl chloride in 20 mL of dichloromethane solution was slowly added and stirred at room temperature for 2 h. After the reaction was completed, the reaction mixture was washed with water (100 mL × 3), and the organic phase was dried over with Na_2_SO_4_ and concentrated under reduced pressure. The crude product was purified by column chromatography using dichloromethane/methanol (V_dichloromethane_/V_methanol_ = 98:2→95:5) as the eluent to give compounds **5a**–**5j**.

*N-(3,4-dimethyl-2-oxo-2H-chromen-7-yl)furan-2-carboxamide (**5a**)*: a gray solid; mp: 270.1–271.6 °C; ^1^H NMR (400 MHz, DMSO-*d*_6_) δ 10.50 (s, 1H), 7.98 (d, *J* = 0.8 Hz, 1H), 7.85 (s, 1H), 7.71 (s, 2H), 7.39 (d, *J* = 3.4 Hz, 1H), 6.73 (q, *J* = 3.4, 1.7 Hz, 1H), 2.34 (s, 3H), 2.06 (s, 3H); ^13^C NMR (101 MHz, DMSO-*d*_6_) δ 161.5, 156.8, 152.2, 147.6, 146.9, 146.6, 141.3, 125.8, 119.8, 116.5, 116.3, 116.0, 112.8, 106.8, 15.2, 13.5; HR-MS (ESI): *m*/*z* calcd for C_16_H_14_NO_4_^+^ ([M + H]^+^) 284.0917, found 284.0924.

*N-(3-ethyl-4-methyl-2-oxo-2H-chromen-7-yl)furan-2-carboxamide (**5b**)*: a gray solid; mp: 239.2–240.8 °C; ^1^H NMR (400 MHz, DMSO-*d*_6_) δ 10.52 (s, 1H), 7.98 (d, *J* = 0.9 Hz, 1H), 7.87 (s, 1H), 7.73 (s, 2H), 7.40 (d, *J* = 3.4 Hz, 1H), 6.74 (dd, *J* = 3.5, 1.7 Hz, 1H), 2.56 (q, *J* = 7.4 Hz, 2H), 2.38 (s, 3H), 1.05 (t, *J* = 7.4 Hz, 3H); ^13^C NMR (101 MHz, DMSO-*d*_6_) δ 161.1, 156.9, 152.4, 147.6, 146.6, 146.6, 141.4, 126.0, 125.6, 116.5, 116.4, 116.0, 112.8, 106.8, 20.8, 14.7, 13.4; HR-MS (ESI): *m*/*z* calcd for C_17_H_16_NO_4_^+^ ([M + H]^+^) 298.1074, found 298.1080.

*N-(6-oxo-7,8,9,10-tetrahydro-6H-benzo[c]chromen-3-yl)furan-2-carboxamide (**5c**)*: a yellow solid; mp: 287.5–289.0 °C; ^1^H NMR (400 MHz, DMSO-*d*_6_) δ 10.53 (s, 1H), 7.98 (s, 1H), 7.88 (s, 1H), 7.73 (d, *J* = 8.3 Hz, 1H), 7.67 (d, *J* = 8.7 Hz, 1H), 7.40 (d, *J* = 3.1 Hz, 1H), 6.74 (s, 1H), 2.77 (s, 2H), 2.41 (s, 2H), 1.87–1.52 (m, 4H); ^13^C NMR (126 MHz, DMSO-*d*_6_) δ 161.2, 157.0, 152.4, 147.9, 147.6, 146.4, 141.3, 124.5, 121.6, 116.7, 116.1, 115.8, 112.7, 107.3, 25.1, 24.2, 21.7, 21.4; HR-MS (ESI): *m*/*z* calcd for C_18_H_16_NO_4_^+^ ([M + H]^+^) 310.1074, found 310.1074.

*N-(3-fluoro-4-methyl-2-oxo-2H-chromen-7-yl)furan-2-carboxamide (**5d**)*: a brown solid; mp: 268.2–270.0 °C; ^1^H NMR (400 MHz, DMSO-*d*_6_) δ 10.56 (s, 1H), 7.98 (s, 1H), 7.93 (d, *J* = 1.5 Hz, 1H), 7.79 (dd, *J* = 8.8, 1.3 Hz, 1H), 7.73 (d, *J* = 8.7 Hz, 1H), 7.40 (d, *J* = 3.4 Hz, 1H), 6.74 (dd, *J* = 3.2, 1.5 Hz, 1H), 2.35 (d, *J* = 2.6 Hz, 3H); ^13^C NMR (101 MHz, DMSO-*d*_6_) δ 156.9, 155.0 (d, *J* = 30.0 Hz), 150.6 (d, *J* = 2.0 Hz), 147.5, 146.7, 141.7, 141.5 (d, *J* = 2.0 Hz), 131.8 (d, *J* = 14.0 Hz), 126.3 (d, *J* = 6.0 Hz), 117.2, 116.1, 115.1 (d, *J* = 3.0 Hz), 112.8, 107.1, 10.3 (d, *J* = 3.0 Hz); HR-MS (ESI): *m*/*z* calcd for C_15_H_11_FNO_4_^+^ ([M + H]^+^) 288.0667, found 288.0669.

*N-(3-chloro-4-methyl-2-oxo-2H-chromen-7-yl)furan-2-carboxamide (**5e**)*: a yellow solid; mp: 300.0–301.7 °C; ^1^H NMR (400 MHz, DMSO-*d*_6_) δ 10.61 (s, 1H), 7.99 (s, 1H), 7.93 (s, 1H), 7.82–7.75 (m, 2H), 7.42 (d, *J* = 3.3 Hz, 1H), 6.74 (d, *J* = 1.6 Hz, 1H), 2.52 (s, 3H); ^13^C NMR (101 MHz, DMSO-*d*_6_) δ 156.9, 156.8, 151.8, 148.9, 147.4, 146.8, 142.6, 126.7, 118.0, 117.0, 116.2, 115.4, 112.9, 106.8, 16.4; HR-MS (ESI): *m*/*z* calcd for C_15_H_10_ClNO_4_^+^ ([M + H]^+^) 304.0371, found 304.0378.

*N-(2-oxo-4-(trifluoromethyl)-2H-chromen-7-yl)furan-2-carboxamide (**5f**)*: a yellow solid; mp: 240.6–242.3 °C; ^1^H NMR (400 MHz, DMSO-*d*_6_) δ 10.72 (s, 1H), 8.03 (d, *J* = 1.8 Hz, 1H), 8.00 (d, *J* = 0.6 Hz, 1H), 7.83 (dd, *J* = 8.9, 1.8 Hz, 1H), 7.69 (d, *J* = 7.7 Hz, 1H), 7.44 (d, *J* = 3.3 Hz, 1H), 6.90 (s, 1H), 6.75 (q, *J* = 3.4, 1.6 Hz, 1H); ^13^C NMR (126 MHz, DMSO-*d*_6_) δ 159.6, 157.6, 155.5, 147.9, 147.5, 144.1, 140.2 (q, *J* = 32.1 Hz), 126.2, 122.8 (q, *J* = 276.1 Hz), 117.8, 117.0, 115.5 (d, *J* = 5.4 Hz), 113.4, 109.5, 108.1; HR-MS (ESI): *m*/*z* calcd for C_15_H_9_F_3_NO_4_^+^ ([M + H]^+^) 324.0478, found 324.0480.

*N-(3-ethyl-4-methyl-2-oxo-2H-chromen-7-yl)benzamide (**5g**)*: a yellow solid; mp: 243.3–244.9 °C; ^1^H NMR (400 MHz, DMSO-*d*_6_) δ 10.61 (s, 1H), 7.98 (d, *J* = 7.3 Hz, 2H), 7.93 (d, *J* = 1.2 Hz, 1H), 7.78 (d, *J* = 8.8 Hz, 1H), 7.76–7.72 (m, 1H), 7.65–7.60 (m, 1H), 7.59–7.53 (m, 2H), 2.58 (q, *J* = 7.4 Hz, 2H), 2.41 (s, 3H), 1.06 (t, *J* = 7.4 Hz, 3H); ^13^C NMR (101 MHz, DMSO-*d*_6_) δ 166.5, 161.2, 152.4, 146.6, 142.0, 135.0, 132.4, 128.9, 128.3, 126.0, 125.5, 116.5, 116.4, 106.8, 20.8, 14.7, 13.4; HR-MS (ESI): *m*/*z* calcd for C_19_H_18_NO_3_^+^ ([M + H]^+^) 308.1281, found 308.1285.

*N-(6-oxo-7,8,9,10-tetrahydro-6H-benzo[c]chromen-3-yl)benzamide (**5h**)*: a yellow solid; mp: 276.8–277.8 °C; ^1^H NMR (400 MHz, DMSO-*d*_6_) δ 10.58 (s, 1H), 7.97 (d, *J* = 7.2 Hz, 2H), 7.91 (d, *J* = 1.8 Hz, 1H), 7.72 (dd, *J* = 8.7, 1.8 Hz, 1H), 7.64 (dd, *J* = 16.6, 8.0 Hz, 2H), 7.56 (t, *J* = 7.4 Hz, 2H), 2.84–2.70 (m, 2H), 2.46–2.35 (m, 2H), 1.82–1.68 (m, 4H); ^13^C NMR (101 MHz, DMSO-*d*_6_) δ 166.4, 161.2, 152.2, 147.6, 141.8, 135.0, 132.4, 128.9, 128.2, 124.6, 121.2, 116.5, 115.8, 106.8, 25.0, 24.1, 21.6, 21.2; HR-MS (ESI): *m*/*z* calcd for C_20_H_18_NO_3_^+^ ([M + H]^+^) 320.1281, found 320.1290.

*N-(3-fluoro-4-methyl-2-oxo-2H-chromen-7-yl)benzamide (**5i**)*: a yellow solid; mp: 278.8–280.3 °C; ^1^H NMR (400 MHz, DMSO-*d*_6_) δ 10.59 (s, 1H), 7.97 (d, *J* = 8.1 Hz, 3H), 7.78 (d, *J* = 8.6 Hz, 1H), 7.69 (d, *J* = 8.7 Hz, 1H), 7.64–7.59 (m, 1H), 7.55 (t, *J* = 7.4 Hz, 2H), 2.32 (d, *J* = 2.2 Hz, 3H); ^13^C NMR (101 MHz, DMSO-*d*_6_) δ 166.4, 154.9 (d, *J* = 28.0 Hz), 150.6, 144.1, 142.1, 141.65, 134.83, 132.39, 131.72 (d, *J* = 13.0 Hz), 128.88, 128.24, 126.07 (d, *J* = 6.0 Hz), 117.08, 114.94, 106.89, 10.22 (d, *J* = 3.0 Hz); HR-MS (ESI): *m*/*z* calcd for C_17_H_13_FNO_3_^+^ ([M + H]^+^) 298.0874, found 298.0880.

*N-(3-chloro-4-methyl-2-oxo-2H-chromen-7-yl)benzamide (**5j**)*: a gray solid; mp: 276.0–278.0 °C; ^1^H NMR (400 MHz, DMSO-*d_6_*) δ 10.71 (s, 1H), 8.00 (d, *J* = 1.9 Hz, 2H), 7.98 (s, 1H), 7.87 (d, *J* = 8.8 Hz, 1H), 7.81 (dd, *J* = 8.8, 1.8 Hz, 1H), 7.64 (t, *J* = 7.3 Hz, 1H), 7.57 (t, *J* = 7.4 Hz, 2H), 2.56 (s, 3H); ^13^C NMR (101 MHz, DMSO-*d_6_*) δ 166.6, 156.8, 151.9, 148.9, 143.1, 134.8, 132.5, 128.9, 128.3, 126.7, 118.0, 117.0, 115.3, 106.8, 16.4; HR-MS (ESI): *m*/*z* calcd for C_17_H_13_ClNO_3_^+^ ([M + H]^+^) 314.0579, found 314.0583.

#### 3.2.3. General Procedure for the Synthesis of the Intermediates **5k**–**5t**

Compound **4** (2.5 mmol) was dissolved in tetrahydrofuran (20.0 mL), and then sodium bicarbonate (10.0 mmol) was added to the solution and cooled to 0 °C. To the mixture, 5.0 mmol of aroyl chloride in 20 mL of tetrahydrofuran solution was slowly added and stirred at room temperature for 2 h. After the reaction was completed, the reaction mixture was washed with water (100 mL × 3), and the organic phase was dried over with Na_2_SO_4_ and concentrated under reduced pressure. The crude product was purified by column chromatography using dichloromethane/methanol (V_dichloromethane_/V_methanol_ = 98:2→95:5) as the eluent to obtain compounds **5k**–**5t**.

*N-(3,4-dimethyl-2-oxo-2H-chromen-7-yl)nicotinamide (**5k**)*: a yellow solid; mp: 270.3–271.3 °C; ^1^H NMR (400 MHz, DMSO-*d_6_*) δ 10.78 (s, 1H), 9.13 (d, *J* = 1.8 Hz, 1H), 8.79 (dd, *J* = 4.8, 1.5 Hz, 1H), 8.39–8.23 (m, 1H), 7.91 (d, *J* = 2.0 Hz, 1H), 7.72 (dd, *J* = 8.8, 2.0 Hz, 1H), 7.60 (dd, *J* = 7.5, 4.8 Hz, 1H), 2.39 (s, 3H), 2.10 (s, 3H); ^13^C NMR (101 MHz, DMSO-*d_6_*) δ 164.9, 161.5, 152.9, 152.3, 149.2, 146.9, 141.5, 136.1, 130.7, 125.9, 124.0, 119.9, 116.5, 116.5, 106.9, 15.2, 13.5; HR-MS (ESI): *m*/*z* calcd for C_17_H_14_N_2_O_3_Na^+^ ([M + Na]^+^) 317.0897, found 317.0914.

*N-(3-ethyl-4-methyl-2-oxo-2H-chromen-7-yl)nicotinamide (**5l**)*: a yellow solid; mp: 275.0–276.9 °C; ^1^H NMR (400 MHz, DMSO-*d_6_*) δ 10.77 (s, 1H), 9.13 (d, *J* = 1.7 Hz, 1H), 8.79 (dd, *J* = 4.7, 1.3 Hz, 1H), 8.41–8.20 (m, 1H), 7.89 (d, *J* = 1.8 Hz, 1H), 7.78 (d, *J* = 8.8 Hz, 1H), 7.71 (dd, *J* = 8.8, 1.9 Hz, 1H), 7.60 (dd, *J* = 7.9, 4.8 Hz, 1H), 2.57 (q, *J* = 7.4 Hz, 2H), 2.40 (s, 3H), 1.06 (t, *J* = 7.4 Hz, 3H); ^13^C NMR (101 MHz, DMSO-*d_6_*) δ 164.9, 161.1, 152.8, 152.4, 149.2, 146.5, 141.6, 136.0, 130.6, 126.1, 125.7, 124.0, 116.6, 116.5, 106.9, 20.8, 14.7, 13.4; HR-MS (ESI): *m*/*z* calcd for C_18_H_17_N_2_O_3_^+^ ([M + H]^+^) 309.1234, found 309.1244.

*N-(6-oxo-7,8,9,10-tetrahydro-6H-benzo[c]chromen-3-yl)nicotinamide (**5m**)*: a gray solid; mp: 284.7–285.4 °C; ^1^H NMR (400 MHz, DMSO-*d_6_*) δ 10.76 (s, 1H), 9.12 (d, *J* = 1.7 Hz, 1H), 8.79 (dd, *J* = 4.8, 1.4 Hz, 1H), 8.39–8.19 (m, 1H), 7.89 (s, 1H), 7.75–7.65 (m, 2H), 7.60 (dd, *J* = 7.8, 4.8 Hz, 1H), 2.94–2.61 (m, 2H), 2.37 (d, *J* = 32.1 Hz, 2H), 2.04–1.52 (m, 4H); ^13^C NMR (101 MHz, DMSO-*d_6_*) δ 164.9, 161.2, 152.8, 152.1, 149.2, 147.5, 141.4, 136.0, 130.7, 124.6, 124.0, 121.4, 116.4, 116.0, 106.9, 25.0, 24.1, 21.6, 21.2; HR-MS (ESI): *m*/*z* calcd for C_19_H_17_N_2_O_3_^+^ ([M + H]^+^) 321.1234, found 321.1252.

*N-(3-chloro-4-methyl-2-oxo-2H-chromen-7-yl)nicotinamide (**5n**)*: a brown solid; mp: 243.7–245.0 °C; ^1^H NMR (400 MHz, DMSO-*d_6_*) δ 10.79 (s, 1H), 9.11 (s, 1H), 8.78 (d, *J* = 3.4 Hz, 1H), 8.29 (d, *J* = 7.5 Hz, 1H), 7.97–7.87 (m, 1H), 7.77 (d, *J* = 8.6 Hz, 1H), 7.71 (d, *J* = 8.6 Hz, 1H), 7.62–7.52 (m, 1H), 2.49 (s, 3H); ^13^C NMR (101 MHz, DMSO-*d_6_*) δ 165.0, 156.7, 152.9, 151.8, 149.2, 148.8, 142.7, 136.1, 130.4, 126.7, 124.0, 118.1, 116.9, 115.5, 106.8, 16.4; HR-MS (ESI): *m*/*z* calcd for C_16_H_12_ClN_2_O_3_^+^ ([M + H]^+^) 337.0350, found 337.0363.

*N-(2-oxo-4-(trifluoromethyl)-2H-chromen-7-yl)nicotinamide (**5o**)*: a yellow solid; mp: 232.3–234.1 °C; ^1^H NMR (400 MHz, DMSO-*d_6_*) δ 10.91 (s, 1H), 9.12 (s, 1H), 8.79 (d, *J* = 4.4 Hz, 1H), 8.31 (d, *J* = 7.7 Hz, 1H), 8.02 (s, 1H), 7.76 (d, *J* = 8.8 Hz, 1H), 7.67 (d, *J* = 8.4 Hz, 1H), 7.63–7.52 (m, 1H), 6.89 (s, 1H); ^13^C NMR (101 MHz, CF_3_COOD) δ 164.0, 162.1, 154.2, 146.2, 144.2 (q, *J* = 34.0 Hz), 143.7, 141.6, 140.5, 133.9, 128.0, 126.6, 122.2, 119.5, 118.7, 111.9, 109.7; HR-MS (ESI): *m*/*z* calcd for C_16_H_10_F_3_N_2_O_3_^+^ ([M + H]^+^) 335.0638, found 335.0656.

*2-chloro-N-(3,4-dimethyl-2-oxo-2H-chromen-7-yl)nicotinamide (**5p**)*: a brown solid; mp: 259.7–260.4 °C; ^1^H NMR (400 MHz, DMSO-*d_6_*) δ 11.04 (s, 1H), 8.57 (d, *J* = 3.3 Hz, 1H), 8.16 (dd, *J* = 18.6, 12.3 Hz, 1H), 7.79 (s, 1H), 7.74 (d, *J* = 8.7 Hz, 1H), 7.67–7.58 (m, 1H), 7.55 (t, *J* = 10.2 Hz, 1H), 2.35 (s, 3H), 2.08 (s, 3H); ^13^C NMR (101 MHz, DMSO-*d_6_*) δ 164.4, 161.3, 152.2, 151.2, 146.9, 146.6, 141.0, 138.8, 133.2, 125.9, 123.6, 120.0, 116.5, 115.8, 106.3, 15.1, 13.4; HR-MS (ESI): *m*/*z* calcd for C_17_H_13_ClN_2_O_3_Na^+^ ([M + Na]^+^) 351.0507, found 351.0522.

*2-chloro-N-(3-ethyl-4-methyl-2-oxo-2H-chromen-7-yl)nicotinamide (**5q**)*: a white solid; mp: 237.5–239.0 °C; ^1^H NMR (400 MHz, DMSO-*d_6_*) δ 11.04 (s, 1H), 8.58 (dd, *J* = 4.8, 1.8 Hz, 1H), 8.15 (dd, *J* = 7.5, 1.8 Hz, 1H), 7.80 (d, *J* = 1.9 Hz, 1H), 7.75 (d, *J* = 8.7 Hz, 1H), 7.61 (dd, *J* = 6.7, 4.0 Hz, 1H), 7.60–7.53 (m, 1H), 2.57 (q, *J* = 7.3 Hz, 2H), 2.39 (s, 3H), 1.06 (t, *J* = 7.4 Hz, 3H); ^13^C NMR (101 MHz, DMSO-*d_6_*) δ 164.4, 161.0, 152.4, 151.3, 146.9, 146.4, 141.2, 138.8, 133.2, 126.2, 125.8, 123.7, 116.7, 115.9, 106.3, 20.8, 14.6, 13.3; HR-MS (ESI): *m*/*z* calcd for C_18_H_16_ClN_2_O_3_^+^ ([M + H]^+^) 343.0844, found 343.0874.

*2-chloro-N-(6-oxo-7,8,9,10-tetrahydro-6H-benzo[c]chromen-3-yl)nicotinamide (**5r**)*: a yellow solid; mp: 275.5–274.7 °C; ^1^H NMR (400 MHz, DMSO-*d_6_*) δ 11.03 (s, 1H), 8.58 (dd, *J* = 4.8, 1.8 Hz, 1H), 8.15 (dd, *J* = 7.5, 1.8 Hz, 1H), 7.78 (d, *J* = 1.6 Hz, 1H), 7.69–7.57 (m, 2H), 7.54 (dd, *J* = 8.7, 1.6 Hz, 1H), 2.72 (s, 2H), 2.39 (s, 2H), 1.74 (dd, *J* = 11.3, 5.9 Hz, 4H); ^13^C NMR (101 MHz, DMSO-*d_6_*) δ 164.4, 161.0, 152.2, 151.3, 147.3, 146.9, 140.9, 138.8, 133.2, 124.7, 123.7, 121.5, 116.1, 115.8, 106.3, 24.9, 24.0, 21.5, 21.2; HR-MS (ESI): *m*/*z* calcd for C_19_H_15_ClN_2_O_3_Na^+^ ([M + Na]^+^) 377.0663, found 377.0683.

*2-chloro-N-(3-chloro-4-methyl-2-oxo-2H-chromen-7-yl)nicotinamide (**5s**)*: a brown solid; mp: 259.7–263.2 °C; ^1^H NMR (400 MHz, DMSO-*d_6_*) δ 11.14 (s, 1H), 8.58 (dd, *J* = 4.7, 1.6 Hz, 1H), 8.16 (dd, *J* = 7.5, 1.6 Hz, 1H), 7.88–7.78 (m, 2H), 7.65–7.57 (m, 2H), 2.53 (s, 3H); ^13^C NMR (101 MHz, DMSO-*d_6_*) δ 164.5, 156.6, 151.9, 151.4, 148.7, 146.9, 142.3, 138.8, 133.0, 126.9, 123.7, 118.3, 116.4, 115.7, 106.4, 16.4; HR-MS (ESI): *m*/*z* calcd for C_16_H_11_Cl_2_N_2_O_3_^+^ ([M + H]^+^) 349.0141, found 349.0145.

*2-chloro-N-(2-oxo-4-(trifluoromethyl)-2H-chromen-7-yl)nicotinamide (**5t**)*: a yellow solid; mp: 240.5–242.0 °C; ^1^H NMR (400 MHz, DMSO-*d_6_*) δ 11.25 (s, 1H), 8.59 (dd, *J* = 4.8, 1.7 Hz, 1H), 8.16 (dd, *J* = 7.5, 1.6 Hz, 1H), 7.95 (d, *J* = 1.3 Hz, 1H), 7.72 (d, *J* = 8.4 Hz, 1H), 7.66 (dd, *J* = 8.9, 1.5 Hz, 1H), 7.62 (dd, *J* = 7.5, 4.9 Hz, 1H), 6.93 (s, 1H); ^13^C NMR (101 MHz, DMSO-*d_6_*) δ 164.7, 158.9, 155.0, 151.4, 147.0, 143.2, 139.6 (q, *J* = 32.4 Hz), 138.8, 132.9, 125.9, 123.6, 122.0 (q, *J* = 276.64 Hz), 116.7, 115.1 (d, *J* = 5.3 Hz), 109.3, 107.1; HR-MS (ESI): *m*/*z* calcd for C_16_H_9_ClF_3_N_2_O_3_^+^ ([M + H]^+^) 369.0248, found 369.0257.

#### 3.2.4. General Procedure for the Synthesis of the Intermediates **6a**–**6r**

Compound **4** (2.5 mmol) was dissolved in tetrahydrofuran (20.0 mL), and then DMAP (0.25 mmol) was added to the solution and cooled to 0 °C. To the mixture, 5.0 mmol of aryl sulfonyl chloride in 20 mL of pyridine solution was slowly added and stirred at room temperature for 2 h. After the reaction was completed, the reaction mixture was washed with water (100 mL × 3), and the organic phase was dried over with Na_2_SO_4_ and concentrated under reduced pressure. The crude product was purified by column chromatography using petroleum ether/ethyl acetate (V_petroleum ether_/V_ethyl acetate_ = 5:1→2:1) as the eluent to produce compounds **6a**–**6r**.

*N-(3,4-dimethyl-2-oxo-2H-chromen-7-yl)benzenesulfonamide (**6a**)*: a yellow solid; mp: 257.0–258.8 °C; ^1^H NMR (400 MHz, DMSO-*d_6_*) δ 10.89 (s, 1H), 7.85 (d, *J* = 7.4 Hz, 2H), 7.68–7.48 (m, 4H), 7.08 (dd, *J* = 8.7, 1.9 Hz, 1H), 7.02 (d, *J* = 1.9 Hz, 1H), 2.28 (s, 3H), 2.03 (s, 3H); ^13^C NMR (101 MHz, DMSO-*d_6_*) δ 161.2, 152.4, 146.6, 140.5, 139.6, 133.7, 129.9, 127.2, 126.5, 119.9, 116.3, 115.2, 105.6, 15.1, 13.4; HR-MS (ESI): *m*/*z* calcd for C_17_H_16_SNO_4_^+^ ([M + H]^+^) 330.0795, found 330.0818.

*N-(3,4-dimethyl-2-oxo-2H-chromen-7-yl)-4-methoxybenzenesulfonamide (**6b**)*: a white solid; mp: 207.2–208.6 °C; ^1^H NMR (400 MHz, DMSO-*d_6_*) δ 10.74 (s, 1H), 7.78 (d, *J* = 8.9 Hz, 2H), 7.62 (d, *J* = 8.7 Hz, 1H), 7.12–7.04 (m, 3H), 7.01 (d, *J* = 1.9 Hz, 1H), 3.79 (s, 3H), 2.28 (s, 3H), 2.03 (s, 3H); ^13^C NMR (101 MHz, DMSO-*d_6_*) δ 163.1, 161.2, 152.4, 146.6, 140.7, 131.1, 129.4, 126.5, 119.8, 116.1, 115.1, 115.0, 105.3, 56.1, 15.1, 13.4; HR-MS (ESI): *m*/*z* calcd for C_18_H_17_SNO_5_Na^+^ ([M + Na]^+^) 382.0720, found 382.0743.

*N-(3,4-dimethyl-2-oxo-2H-chromen-7-yl)-4-fluorobenzenesulfonamide (**6c**)*: a yellow solid; mp: 238.5–240.0 °C; ^1^H NMR (400 MHz, DMSO-*d_6_*) δ 10.92 (s, 1H), 7.91 (dd, *J* = 8.6, 5.1 Hz, 2H), 7.50 (d, *J* = 8.7 Hz, 1H), 7.39 (t, *J* = 8.7 Hz, 2H), 7.03 (d, *J* = 8.7 Hz, 1H), 6.99 (s, 1H), 2.15 (s, 3H), 1.93 (s, 3H); ^13^C NMR (101 MHz, DMSO-*d_6_*) δ 166.2, 163.7, 161.1, 152.4, 146.4, 140.2, 136.0 (d, *J* = 3.0 Hz), 130.3 (d, *J* = 10.0 Hz), 126.5, 120.0, 117.1 (d, *J* = 23.0 Hz), 115.9 (d, *J* = 107.53 Hz), 105.8, 15.0, 13.3; HR-MS (ESI): *m*/*z* calcd for C_17_H_14_SFNO_4_Na^+^ ([M + Na]^+^) 370.0520, found 370.0539.

*N-(3,4-dimethyl-2-oxo-2H-chromen-7-yl)-4-chlorobenzenesulfonamide (**6d**)*: a white solid; mp: 230.7–231.2 °C; ^1^H NMR (400 MHz, DMSO-*d_6_*) δ 10.97 (s, 1H), 7.84 (d, *J* = 8.5 Hz, 2H), 7.64 (t, *J* = 8.9 Hz, 3H), 7.16–6.93 (m, 2H), 2.27 (s, 3H), 2.02 (s, 3H); ^13^C NMR (101 MHz, DMSO-*d_6_*) δ 161.2, 152.4, 146.7, 140.2, 138.7, 138.4, 130.2, 129.1, 126.7, 120.1, 116.6, 115.5, 105.9, 15.2, 13.4; HR-MS (ESI): *m*/*z* calcd for C_17_H_15_SClNO_4_^+^ ([M + H]^+^) 364.0405, found 364.0429.

*N-(3,4-dimethyl-2-oxo-2H-chromen-7-yl)-4-bromobenzenesulfonamide (**6e**)*: a yellow solid; mp: 238.4–240.1 °C; ^1^H NMR (400 MHz, DMSO-*d_6_*) δ 10.95 (s, 1H), 7.80 (d, *J* = 8.7 Hz, 2H), 7.75 (d, *J* = 8.6 Hz, 2H), 7.65 (d, *J* = 8.7 Hz, 1H), 7.07 (dd, *J* = 8.7, 2.1 Hz, 1H), 7.03 (d, *J* = 2.0 Hz, 1H), 2.29 (s, 3H), 2.04 (s, 3H); ^13^C NMR (101 MHz, DMSO-*d_6_*) δ 161.2, 152.4, 146.6, 140.1, 138.8, 133.1, 129.2, 127.7, 126.7, 120.1, 116.6, 115.4, 105.9, 15.1, 13.4; HR-MS (ESI): *m*/*z* calcd for C_17_H_14_SBrNO_4_Na^+^ ([M + Na]^+^) 429.9719, found 429.9743.

*N-(3-ethyl-4-methyl-2-oxo-2H-chromen-7-yl)benzenesulfonamide (**6f**)*: a gray solid; mp: 199.9–200.4 °C; ^1^H NMR (400 MHz, DMSO-*d_6_*) δ 10.92 (s, 1H), 7.89–7.81 (m, 2H), 7.64–7.62 (m, 2H), 7.60–7.56 (m, 2H), 7.08 (dd, *J* = 8.7, 2.2 Hz, 1H), 7.04 (dd, *J* = 8.8, 2.2 Hz, 1H), 2.52–2.50 (m, 2H), 2.30 (s, 3H), 0.99 (t, *J* = 7.4 Hz, 3H); ^13^C NMR (101 MHz, DMSO-*d_6_*) δ 160.8, 152.5, 146.5, 140.6, 139.6, 133.8, 130.0, 127.2, 126.9, 125.7, 116.5, 115.3, 105.6, 20.7, 14.6, 13.3; HR-MS (ESI): *m*/*z* calcd for C_18_H_17_SNO_4_Na^+^ ([M + Na]^+^) 366.0771, found 366.0794.

*N-(3-ethyl-4-methyl-2-oxo-2H-chromen-7-yl)-4-methoxybenzenesulfonamide (**6g**)*: a white solid; mp: 204.4–205.6 °C; ^1^H NMR (400 MHz, DMSO-*d_6_*) δ 10.77 (s, 1H), 7.78 (d, *J* = 8.8 Hz, 2H), 7.63 (d, *J* = 8.7 Hz, 1H), 7.12–7.05 (m, 3H), 7.02 (t, *J* = 5.3 Hz, 1H), 3.78 (s, 3H), 2.51 (q, *J* = 6.0 Hz, 2H), 2.30 (s, 3H), 1.00 (t, *J* = 7.4 Hz, 3H); ^13^C NMR (101 MHz, DMSO-*d_6_*) δ 163.1, 160.8, 152.5, 146.4, 140.8, 131.1, 129.4, 126.7, 125.6, 116.2, 115.1, 115.0, 105.3, 56.1, 20.7, 14.6, 13.3; HR-MS (ESI): *m*/*z* calcd for C_19_H_19_SNO_5_Na^+^ ([M + Na]^+^) 396.0876, found 396.0907.

*N-(3-ethyl-4-methyl-2-oxo-2H-chromen-7-yl)-4-fluorobenzenesulfonamide (**6h**)*: a gray solid; mp: 209.8–211.3 °C; ^1^H NMR (400 MHz, DMSO-*d_6_*) δ 10.93 (s, 1H), 7.98–7.79 (m, 2H), 7.65 (d, *J* = 8.7 Hz, 1H), 7.42 (t, *J* = 8.8 Hz, 2H), 7.07 (dd, *J* = 8.7, 2.1 Hz, 1H), 7.03 (d, *J* = 2.1 Hz, 1H), 2.52 (d, *J* = 7.6 Hz, 2H), 2.31 (s, 3H), 1.00 (t, *J* = 7.4 Hz, 3H); ^13^C NMR (101 MHz, DMSO-*d_6_*) δ 166.2, 163.7, 160.7, 152.5, 146.3, 140.4, 135.9 (d, *J* = 3.0Hz), 130.4 (d, *J* = 10.0Hz), 126.3 (d, *J* = 97.0Hz), 117.2 (d, *J* = 23.0Hz), 116.5, 115.4, 105.8, 20.7, 14.5, 13.2; HR-MS (ESI): *m*/*z* calcd for C_18_H_16_SFNO_4_Na^+^ ([M + Na]^+^) 384.0676, found 384.0709.

*N-(3-ethyl-4-methyl-2-oxo-2H-chromen-7-yl)-4-chlorobenzenesulfonamide (**6i**)*: a white solid; mp: 214.6–216.2 °C; ^1^H NMR (400 MHz, DMSO-*d_6_*) δ 10.93 (s, 1H), 7.98–7.79 (m, 2H), 7.65 (d, *J* = 8.7 Hz, 1H), 7.42 (t, *J* = 8.8 Hz, 2H), 7.07 (dd, *J* = 8.7, 2.1 Hz, 1H), 7.03 (d, *J* = 2.1 Hz, 1H), 2.52 (d, *J* = 7.6 Hz, 2H), 2.31 (s, 3H), 1.00 (t, *J* = 7.4 Hz, 3H); ^13^C NMR (101 MHz, DMSO-*d_6_*) δ 160.7, 152.5, 146.3, 140.2, 138.7, 138.4, 130.1, 129.1, 126.8, 125.9, 116.7, 115.5, 105.9, 20.7, 14.6, 13.2; HR-MS (ESI): *m*/*z* calcd for C_18_H_17_SClNO_4_^+^ ([M + H]^+^) 378.0561, found 378.0592.

*N-(3-ethyl-4-methyl-2-oxo-2H-chromen-7-yl)-4-bromobenzenesulfonamide (**6j**)*: an orange solid; mp: 215.7–217.0 °C; ^1^H NMR (400 MHz, DMSO-*d_6_*) δ 10.98 (s, 1H), 7.85–7.69 (m, 4H), 7.60 (d, *J* = 8.7 Hz, 1H), 7.06 (dd, *J* = 8.7, 1.9 Hz, 1H), 7.02 (d, *J* = 1.8 Hz, 1H), 2.52–2.41 (m, 2H), 2.26 (s, 3H), 0.97 (t, *J* = 7.4 Hz, 3H); ^13^C NMR (101 MHz, DMSO-*d_6_*) δ 160.7, 152.5, 146.3, 140.2, 138.8, 133.1, 129.2, 127.7, 126.9, 125.9, 116.7, 115.4, 105.9, 20.7, 14.6, 13.3; HR-MS (ESI): *m*/*z* calcd for C_20_H_20_SBrNO_4_^+^ ([M + H]^+^) 422.0056, found 422.0053.

*N-(6-oxo-7,8,9,10-tetrahydro-6H-benzo[c]chromen-3-yl)-4-methoxybenzenesulfonamide (**6k**)*: a white solid; mp: 218.5–220.1 °C; ^1^H NMR (400 MHz, DMSO-*d_6_*) δ 10.72 (s, 1H), 7.77 (d, *J* = 8.9 Hz, 2H), 7.56 (d, *J* = 8.6 Hz, 1H), 7.07 (dd, *J* = 10.9, 5.3 Hz, 3H), 7.02 (d, *J* = 1.9 Hz, 1H), 3.79 (s, 3H), 2.68 (d, *J* = 5.5 Hz, 2H), 2.37 (s, 2H), 1.73–1.69 (m, 4H); ^13^C NMR (101 MHz, DMSO-*d_6_*) δ 163.1, 160.8, 152.3, 147.4, 140.6, 131.1, 129.4, 125.3, 121.3, 115.7, 115.1, 115.0, 105.5, 56.1, 24.9, 24.0, 21.5, 21.1; HR-MS (ESI): *m*/*z* calcd for C_20_H_20_SNO_5_^+^ ([M + H]^+^) 386.1056, found 386.1052.

*N-(6-oxo-7,8,9,10-tetrahydro-6H-benzo[c]chromen-3-yl)-4-fluorobenzenesulfonamide (**6l**)*: a yellow solid; mp: 215.8–217.0 °C; ^1^H NMR (400 MHz, DMSO-*d_6_*) δ 10.89 (s, 1H), 7.89 (ddd, *J* = 8.1, 5.1, 2.5 Hz, 2H), 7.58 (d, *J* = 8.6 Hz, 1H), 7.42 (dd, *J* = 12.3, 5.4 Hz, 2H), 7.07 (dd, *J* = 8.6, 2.1 Hz, 1H), 7.04 (d, *J* = 2.1 Hz, 1H), 2.69 (s, 2H), 2.37 (s, 2H), 1.83–1.54 (m, 4H); ^13^C NMR (101 MHz, DMSO-*d_6_*) δ 166.2, 163.7, 160.8, 152.3, 147.4, 140.1, 135.9 (d, *J* = 3.0 Hz), 130.3 (d, *J* = 9.0 Hz), 125.4, 121.6, 117.2 (d, *J* = 23.0 Hz), 115.7 (d, *J* = 58.0 Hz), 105.9, 24.9, 24.0, 21.5, 21.1; HR-MS (ESI): *m*/*z* calcd for C_19_H_17_SFNO_4_^+^ ([M + H]^+^) 374.0856, found 374.0855.

*N-(6-oxo-7,8,9,10-tetrahydro-6H-benzo[c]chromen-3-yl)-4-bromobenzenesulfonamide (**6m**)*: a pink solid; mp: 217.7–219.3 °C; ^1^H NMR (400 MHz, DMSO-*d_6_*) δ 10.95 (s, 1H), 7.86–7.66 (m, 4H), 7.48 (d, *J* = 8.5 Hz, 1H), 7.10–6.97 (m, 2H), 2.60 (s, 2H), 2.32 (s, 2H), 1.66 (d, *J* = 4.3 Hz, 4H); ^13^C NMR (101 MHz, DMSO-*d_6_*) δ 160.8, 152.3, 147.4, 140.0, 138.8, 133.1, 129.2, 127.7, 125.5, 121.7, 116.1, 115.5, 106.0, 24.9, 24.0, 21.5, 21.1; HR-MS (ESI): *m*/*z* calcd for C_19_H_17_SBrNO_4_^+^ ([M + H]^+^) 434.0056, found 434.0052.

*N-(3-chloro-4-methyl-2-oxo-2H-chromen-7-yl)benzenesulfonamide (**6n**)*: a yellow solid; mp: 260.2–261.8 °C; ^1^H NMR (400 MHz, DMSO-*d_6_*) δ 11.09 (s, 1H), 7.94–7.80 (m, 2H), 7.67–7.61 (m, 2H), 7.61–7.54 (m, 2H), 7.12 (dt, *J* = 15.1, 7.6 Hz, 1H), 7.06 (d, *J* = 2.1 Hz, 1H), 2.40 (s, 3H); ^13^C NMR (126 MHz, DMSO-*d_6_*) δ 157.0, 152.5, 149.2, 142.2, 140.0, 134.4, 130.5, 128.0, 127.7, 118.6, 116.0, 115.9, 105.8, 16.8; HR-MS (ESI): *m*/*z* calcd for C_16_H_13_SClNO_4_^+^ ([M + H]^+^) 350.0248, found 350.0246.

*N-(3-chloro-4-methyl-2-oxo-2H-chromen-7-yl)-4-methoxybenzenesulfonamide (**6o**)*: a yellow solid; mp: 242.9–244.2 °C; ^1^H NMR (400 MHz, DMSO-*d_6_*) δ 10.93 (s, 1H), 7.80 (d, *J* = 8.9 Hz, 1H), 7.73 (d, *J* = 8.8 Hz, 1H), 7.16–7.12 (m, 1H), 7.10 (d, *J* = 8.9 Hz, 1H), 7.07 (d, *J* = 2.0 Hz, 1H), 3.80 (s, 1H), 2.47 (s, 1H); ^13^C NMR (101 MHz, DMSO-*d_6_*) δ 163.2, 156.5, 152.1, 148.7, 142.0, 131.0, 129.5, 127.4, 118.0, 115.3, 115.2, 115.1, 105.1, 56.1, 16.3; HR-MS (ESI): *m*/*z* calcd for C_17_H_15_SClNO_5_^+^ ([M + H]^+^) 380.0354, found 380.0360.

*N-(3-chloro-4-methyl-2-oxo-2H-chromen-7-yl)-4-fluorobenzenesulfonamide (**6p**)*: a yellow solid; mp: 234.4–235.1 °C; ^1^H NMR (400 MHz, DMSO-*d_6_*) δ 11.09 (s, 1H), 7.97–7.88 (m, 2H), 7.65 (d, *J* = 8.7 Hz, 1H), 7.41 (t, *J* = 8.8 Hz, 2H), 7.11 (dd, *J* = 8.7, 2.0 Hz, 1H), 7.05 (d, *J* = 2.0 Hz, 1H), 2.39 (s, 3H); ^13^C NMR (101 MHz, DMSO-*d_6_*) δ 166.3, 163.8, 156.4, 152.0, 148.6, 141.5, 135.9 (d, *J* = 3.0 Hz), 130.3 (d, *J* = 9.0 Hz), 127.4, 118.2, 117.2 (d, *J* = 23.0 Hz), 115.5 (d, *J* = 10.0 Hz), 105.5, 16.3; HR-MS (ESI): *m*/*z* calcd for C_16_H_12_SFClNO_4_^+^ ([M + H]^+^) 368.0154, found 368.0153.

*N-(3-chloro-4-methyl-2-oxo-2H-chromen-7-yl)-4-chlorobenzenesulfonamide (**6q**)*: a yellow solid; mp: 220.1–221.8 °C; ^1^H NMR (400 MHz, DMSO-*d_6_*) δ 11.13 (s, 1H), 7.87 (d, *J* = 8.6 Hz, 2H), 7.75 (d, *J* = 8.7 Hz, 1H), 7.67 (d, *J* = 8.6 Hz, 2H), 7.14 (dd, *J* = 8.7, 2.1 Hz, 1H), 7.09 (d, *J* = 2.0 Hz, 1H), 2.48 (s, 3H); ^13^C NMR (101 MHz, DMSO-*d_6_*) δ 156.5, 152.1, 148.7, 141.4, 138.8, 138.3, 130.2, 129.1, 127.6, 118.3, 115.7, 115.7, 105.7, 16.4; HR-MS (ESI): *m*/*z* calcd for C_16_H_12_SCl_2_NO_4_^+^ ([M + H]^+^) 383.9858, found 383.9842.

*N-(3-chloro-4-methyl-2-oxo-2H-chromen-7-yl)-4-bromobenzenesulfonamide (**6r**)*: a yellow solid; mp: 207.8–209.2 °C; ^1^H NMR (400 MHz, DMSO-*d_6_*) δ 11.13 (s, 1H), 7.82 (d, *J* = 8.8 Hz, 1H), 7.78 (d, *J* = 8.8 Hz, 1H), 7.74 (d, *J* = 8.7 Hz, 1H), 7.14 (dd, *J* = 8.7, 1.9 Hz, 1H), 7.09 (d, *J* = 1.9 Hz, 1H), 2.47 (s, 2H); ^13^C NMR (101 MHz, DMSO-*d_6_*) δ 156.5, 152.1, 148.7, 141.4, 138.7, 133.2, 129.2, 127.9, 127.7, 118.3, 115.7, 105.6, 16.4; HR-MS (ESI): *m*/*z* calcd for C_16_H_12_SClBrNO_4_^+^ ([M + H]^+^) 429.9331, found 429.9325.

### 3.3. Antifungal Bioassay

The antifungal activity of the target compounds against *Botrytis cinerea*, *Alternaria solani*, *Gibberella zeae*, *Rhizoctorzia solani*, *Cucumber anthrax*, and *Alternaria* leaf spot was assessed by the mycelium growth rate method according to the literature [26]. The tested compounds (10 mg) were dissolved in 2 mL of DMSO to make a solution of 5 mg/mL. The solution (0.1 mL) was taken and added to 50 mL of sterilized PDA medium to make a drug-containing medium with a concentration of 50 μg/mL. The medium was poured into 3 sterile petri dishes with an average diameter of 9 cm. The medium with the same amount of DMSO (0.1 mL) added was used as a control. Osthole was taken as the positive control at the same concentration. The preserved strains could be used after being activated twice continuously in a fresh sterile PDA medium. A puncher (inner diameter of 0.5 cm) was used to make a bacterial cake at the edge of the colony. The bacterial cake was inserted into the center of the medium plate with an inoculation needle and cultured in an incubator at 25 °C. The diameters of the colonies on the medium for solvent control were measured when they grew to 2/3 of the diameter of the plate. The diameters of each colony were measured twice with the cross method, and the average value was calculated. Each concentration and two controls were repeated three times, and the data were averaged. According to the preliminary screening results of in vitro activity, compounds with an inhibition rate greater than 60% at a concentration of 50 μg/mL were further tested for their EC_50_ values. The concentration gradients of 50, 25, 12.5, 6.25, and 3.125 μg/mL were set, and the EC_50_ values of the selected compounds against the tested plant pathogenic fungi were determined by the mycelial growth rate method. The data are listed in Table 3 and Table 4, respectively.

## 4. Conclusions

In summary, two series of coumarin 7-carboxamide/sulfonamide derivatives were designed and efficiently synthesized. All target compounds were confirmed by ^1^H NMR,^13^C NMR, and HRMS spectra. Biological assays indicated that the synthesized coumarin derivatives the antifungal activity displayed against *Botrytis cinerea* and *Rhizoctorzia solani* was generally better than that against *Alternaria solani*, *Gibberella zeae*, *Cucumber anthrax*, and *Alternaria* leaf spot. In particular, compounds **6d**, **6e**, **6q**, and **6r** possessed effective antifungal activity against *Botrytis cinerea* (EC_50_ = 27.34, 27.76, 27.78, and 20.52 µg/mL), which was better than that of Osthole (33.67 μg/mL). Among them, compound **6r** was the most promising candidates for further study. Further investigations to modify the coumarin derivatives are well underway, aiming to improve their levels of antifungal activity.

## Data Availability

The data presented in this study are available in the article and Appendix A.

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
