# Peer review of "Novel Coumarin 7-Carboxamide/Sulfonamide Derivatives as Potential Fungicidal Agents: Design, Synthesis, and Biological Evaluation"

_molecules, 2022, doi:10.3390/molecules27206904_

Round 1

Reviewer 1 Report

In the present manuscript, Zhang and co-workers reported the synthesis and antifungal properties evaluation of various coumarin derivatives bearing an amide (or sulfonamide) group at position 7 of the heterocyclic core. The work (topic and results) is interesting for synthetic organic and medicinal chemistry chemists; the synthetic method is straightforward and practical, and the products are well characterized. However, these results are not surprising given the trivial nature of the described reactions and because the scope of the substrate seems limited (several products but little structural diversity). In addition, the manuscript has various spelling and writing mistakes.

Consequently, I believe this paper might be appropriate for its publication in Molecules journal after major revisions and answers to the following questions, suggestions, and/or corrections:

Introduction:

-The conceptualization seems good; however, it is a bit extensive, and additionally, relevant synthetic details should be included.

Discussion:

-It is worth delving a little deeper into the synthetic and bioactivity discussion. In fact, what is discussed about NMR and HRMS analysis is unnecessary; this is already an explicit part of the experimental section and should have been better discussed in the reactions part.

Conclusions:

-The conclusions should be made more precise.

Materials and Methods:

-The + sign (in superscript) must be entered for all molecular formulas because they represent the respective cation as M + H+ (or M + Na+); i.e., ‘C16H14NO4’ must be ‘C16H14NO4+’.

-The 13C NMR data must be reported with a single decimal figure, and the type of carbon should be specified as C, CH, CH2, or CH3.

Supporting Information:

-The authors must place in the Supporting Information the general procedure for synthesizing 7-aminocoumarins and, at this point, cite the respective articles.

-Section 3 in Supporting Information ' Characterization Data of Products' should be deleted because it already appears in the main manuscript and is redundant.

-The authors should include some expansions of the NMR spectra to visualize the respective signals and multiplicity better.

-What happened to the HRMS spectra? The authors should introduce these.

General.

Abstract:

-Change ‘century and dozens’ to ‘century, and dozens’

-Change ‘In this study, we focused on the introduction amide and sulfonamide moieties on the coumarin core, to...’ to ‘This study focused on introducing amide and sulfonamide moieties in the coumarin core to...’

-Change ‘Based on this strategy in mind,’ to ‘Based on this strategy,’

-Change ‘derivatives and their fungicidal’ to ‘derivatives, and their fungicidal’

-Change ‘assays, and this is highlighted by compounds 6r.’ to ‘assays, highlighted by compound 6r.’

-Change ‘by compounds 6r. Among them, compounds 6r’ to ‘by compound 6r. Compound 6r exhibited’

-Change ‘(EC50 = 20.52 µg/mL), and’ to ‘(EC50 = 20.52 µg/mL) and’

Introduction

-Change ‘crisis, and about 2 / 3 of the losses are caused by plant pathogenic fungi and bacteria.’ to ‘crises, and plant pathogenic fungi and bacteria cause about 2 / 3 losses.’

-Change ‘the comprehensive control of agricultural’ to ‘comprehensively controlling agricultural’

-Change ‘Therefore, the development of new,’ to ‘Therefore, developing new,’

-Change ‘friendly, efficient and selective’ to ‘friendly, efficient, and selective’

-Change ‘food, crops and ecological’ to ‘food, crops, and the ecological’

-Change ‘action and environmental’ to ‘action, and environmental’

-Change ‘activity, and have been found widespread’ to ‘activity. They have been found in widespread’

-Change ‘activity and structural plasticity.’ to ‘activity, and structural plasticity.’

-Change ‘They are considered to be a class’ to ‘They are a class’

-Change ‘for the synthesis of small molecules’ to ‘for synthesizing small molecules’

-Change ‘has good inhibitory effect’ to ‘has a good inhibitory effect’

-Change ‘capsici, and also has good inhibitory’ to ‘capsici and has a good inhibitory’

-Change ‘many plant dieases’ to ‘many plant diseases’

-Change ‘mildew with good’ to ‘mildew with a good’

-Change ‘Inspired by these facts, Osthole was employed as a lead structure, our group designed and synthesized series of coumarin derivatives’ to ‘Inspired by these facts and using Osthole as a lead structure, our group designed and synthesized a series of coumarin derivatives’

-Change ‘activity, such as’ to ‘activity. Compounds such as’

-Change ‘ubstituted coumairn derivatives’ to ‘ubstituted coumarin derivatives’

-Change ‘oxime ether moiety.’ to ‘oxime ether moiety were studied.’

-Change ‘agricultural fungicide still’ to ‘agricultural fungicides still’

-Change ‘Novel varieties of commercial amide fungicides have been successively developed by Bayer, BASF, Syngenta and other companies, such as sopyrazam, fluopyram and bixafen’ to ‘Bayer has successively developed novel varieties of commercial amide fungicides, BASF, Syngenta, and other companies, such as sopyrazam, fluopyram, and bixafen’

-Change ‘used in the field of pesticides. Amisulbrom applied to control of phytophthora’ to ‘used in pesticides. Amisulbrom was applied to control phytophthora’

-Change ‘that the combination of the coumarin skeleton with amide and sulfonamides moieties to ‘that combining the coumarin skeleton with amide and sulfonamide moieties’

-Change ‘six phytopathogenic fungi, and with the aim to further improve their fungicidal effects.’ to ‘six phytopathogenic fungi to further improve their fungicide effects.’

Results and discussion

-Change ‘Six kinds of 7-aminocoumarin derivatives were obtained by the reported route,’ to ‘The reported route led to six kinds of 7-aminocoumarin derivatives,’

-Change ‘Five kinds of aroyl chloride’ to ‘Five types of aroyl chloride’

-Change ‘good yields, when CH2Cl2 as the solvent, triethylamine as the base, as shown in Scheme 1.’ to ‘good yields in CH2Cl2 as a solvent and triethylamine as a base (Scheme 1).’

-Change ‘under this reaction conditions, when THF as the solvent, NaHCO3 as the base, the compounds 5k-5v’ to ‘under these reaction conditions; when THF was the solvent and NaHCO3 was the base, the compounds 5k-5v’

-Change ‘and compounds 6a-6r’ to ‘and the compounds 6a-6r’

-Change ‘molecules listed in Table 1 and Table 2, all of them have been confirmed by HRMS, 1H NMR and 13C NMR’ to ‘molecules listed in Tables 1 and 2 have been confirmed by HRMS, 1H NMR, and 13C NMR.’

-Change ‘1H NMR peaks of 10.58 ppm’ to ‘1H NMR signal to 10.58 ppm’

-Change ‘was assigned to the signals of H at C-7 site in amide group, 2.34 and 2.06 ppm were assigned to the signals of H at C-3 and C-2 sites in methyl group, respectively.’ to ‘was assigned to the NH of the amide group. At 2.34 and 2.06 ppm, the signals corresponding to the methyl groups at positions 4 and 3 appear.’

-Change ‘a signal peak at 162.51ppm’ to ‘the signal at 162.5 ppm’

-Change ‘was assigned to carbonyl group in amide group’ to ‘was assigned to the carbonyl group in the amide moiety’

-Change ‘of carbonyl group in coumarin motif.’ to ‘of the carbonyl group in the coumarin motif’

-Change ‘were assigned to –CH3 group’ to ‘were assigned to the methyl groups.’

-Change ‘for C16H13NO4 [M + H]+’ to ‘for C16H13NO4+ [M + H]+

-Change ‘method and the assay results were shown’ to ‘method, and the assay results are shown’

-Change ‘adopted as’ to ‘was adopted as’

-Change ‘from department of pathology, college of plant protection, nanjing agricultural university.’ to ‘from the department of pathology, the college of plant protection, Nanjing agricultural university.’

-Change ‘using different concentration by’ to ‘using different concentrations by’

-Change ‘compounds 6d6e6q and 6r showed’ to ‘compounds 6d6e6q, and 6r showed 

-Change ‘of 71.1, 68.2, 78.2 and 81.6%, respectively,’ to ‘of 71.1, 68.2, 78.2, and 81.6%, respectively,’

-Change ‘Botrytis cinerea, with the corresponding’ to ‘Botrytis cinerea, corresponding

-Change ‘In order to comprehensively’ to ‘To comprehensively’

-Change ‘(6d6e6q and 6r)’ to ‘(6d6e6q, and 6r)

-Change ‘activity, the EC50 value’ to ‘activity; the EC50 value’

-Change ‘activity of most of coumarin’ to ‘activity of most coumarin’

-Change ‘has been proven to be certain poor,’ to ‘has been poor,’

-Change ‘structure–activity relationship’ to ‘structure-activity relationship’

-Change ‘activity of sulfonamide group’ to ‘activity of thr sulfonamide group’

-Change ‘helpful, compared with amide group’ to ‘helpful compared with the amide group’

-Change ‘compounds 6d6e6q and 6r exhibited’ to ‘compounds 6d6e6q, and 6r exhibited

-Change ‘at C-3 cite in coumarin core, meanwhile, … ring were helpful …the antifugal activity’ to ‘at the C-3 site in the coumarin core; meanwhile, … ring was helpful … the antifungal activity’.

Reviewer 2 Report

The authors synthesized some carboxamide and sulfonamide derivatives containing a substituted coumarin group and preliminarily evaluated their fungicidal activity against six phytopathogenic fungi. Several target compounds showed visible activity against Botrytis cinerea. The compound 6r exhibited the fungicidal activity against Botrytis cinerea with an EC50 value of 20.52 µg/mL. The following comments are given for the author's reference.

1. It is suggested to change the word “amide” in the title and text to “carboxamide”.

2. It is suggested to clarify the reason why R3 in the molecular structures of target compounds 5a-5v were selected as furan, benzene ring and pyridine, and why R4 in the molecular structures of target compounds 6a-6r were selected at the 4-position of benzene ring?.

3. It was mentioned on lines 121 and 129 that the fungicidal activity was measured at the concentration of 50 µg/mL, but it was said on line 458 that the concentration of the prepared drug-containing medium was 10 µg/mL. What's going on here?

4. The overall fungicidal activity of the target compounds was relatively low. The molecular structures of the target compounds were not particularly novel. The author should discuss how to optimize the structures to improve the bioactivity.

5. The manuscript need to be carefully checked and revised. Such as, compounds 6r on line 16 should changed to singular; "Fusarium graminearum" and "Phytophthora capitici" on line 45, and the scientific names of the pathogens in the header of Table 3 should be italicized; The first letter of the first word on line 163 should be capitalized;  on line 480 should be changed to ,; and so on.

Round 2

Reviewer 1 Report

The revised manuscript delivered most comments in the right way. Now it's good enough to be accepted for publication in Molecules journal.

Reviewer 2 Report

This paper is recommended to be published.